# The Use of Waste Hazelnut Shells as a Reinforcement in the Development of Green Biocomposites

**DOI:** 10.3390/polym14112151

**Published:** 2022-05-25

**Authors:** Manuela Ceraulo, Francesco Paolo La Mantia, Maria Chiara Mistretta, Vincenzo Titone

**Affiliations:** 1Department of Engineering, University of Palermo, VialedelleScienze, 90128 Palermo, Italy; francescopaolo.lamantia@unipa.it (F.P.L.M.); mc.mistretta@unipa.it (M.C.M.); vincenzo_titone1992@libero.it (V.T.); 2INSTM, Consortium for Materials Science and Technology, Via Giusti 9, 50125 Florence, Italy; 3Irritec S.p.A., Via Industriale sn, 98070 Rocca di Caprileone, Italy

**Keywords:** biodegradable polymers, biocomposites, hazelnut shells, mechanical properties, dynamic mechanical analysis (DMA), rheology

## Abstract

Biodegradable Mater-Bi (MB) composites reinforced with hazelnut shell (HS) powder were prepared in a co-rotating twin-screw extruder followed by compression molding and injection molding. The effects of reinforcement on the morphology, static and dynamic mechanical properties, and thermal and rheological properties of MB/HS biocomposites were studied. Rheological tests showed that the incorporation of HS significantly increased the viscosity of composites with non-Newtonian behavior at low frequencies. On the other hand, a scanning electron microscope (SEM) examination revealed poor interfacial adhesion between the matrix and the filler. The thermal property results indicated that HS could act as a nucleating agent to promote the crystallization properties of biocomposites. Furthermore, the experimental results indicated that the addition of HS led to a significant improvement in the thermomechanical stability of the composites. This paper demonstrates that the incorporation of a low-cost waste product, such as hazelnut shells, is a practical way to produce low-cost biocomposites with good properties. With a content of HS of 10%, a remarkable improvement in the elastic modulus and impact strength was observed in both compression and injection-molded samples. With a higher content of HS, however, the processability in injection molding was strongly worsened.

## 1. Introduction

The unstoppable increase in demand for plastics and the resulting pollution have become a major concern in recent times. [1,2] In fact, recent legislation on material recyclability and environmental requirements are increasingly pushing companies and research institutions to develop environmentally friendly alternatives to stop pollution [3,4,5].

With this in mind, those that are leading the way are biodegradable polymers [6,7] which have recently become more popular and have shown improved quality and functionality, resulting in applicability in food packaging, [8,9,10] agriculture [11,12], and various other applications. [13,14] However, to date, the main disadvantages of most biodegradable polymers are their thermal resistance [15,16] and mechanical properties [17,18], which are inferior to those of conventional polymers. Therefore, recently, a very convenient solution to overcome or at least minimize the low ductility and toughness of biodegradables is biocomposites [19,20,21,22,23].

Biocomposites are a new generation of materials that are critical to the development of various industries to achieve sustainable development goals. Sustainable development could also benefit the hazelnut industry. The hazelnut industry produces a large number of by-products. Of these, waste hazelnut shells represent a potential low-cost reinforcement resource for composite products.

Currently, hazelnut waste has a very low commercial value and is often used either as a heating source or as a raw material to produce furfural in the dye industry [24,25].

In this study, the objectives were to valorize hazelnut waste, particularly HS; provide evidence of its applicability as a reinforcing material for biodegradable polymers; and propose a concrete solution to the problem of HS waste disposal.

The literature contains several studies on organic fillers and biodegradable polymers [26,27]; however, to the best of our knowledge, there are few papers in the literature reporting studies on polymer biocomposites using HS.

Balart et al. [28] used Corylus avellana HS obtained as a by-product of the food industry and evaluated their effects on the mechanical, thermal, and thermo-mechanical properties of PLA composites. The results of their research show that HSs have a slight nucleation effect on poly(lactic acid) chains, leading to an increase in crystallinity, which has a positive effect on the dimensional stability of PLA/HS composites. In another work, the same authors [29] studied the same properties by adding different amounts of epoxidized linseed oil (ELO) to PLA/HS composites to provide a plasticizing effect and improve the inherent low ductility of PLA/HS composites. They reported that ELO had a dual role as a coupling agent and/or a plasticizer, but overall, it improved the overall performance of the PLA/HS composite. Similarly, Aliotta et al. [30] used analytical models to evaluate the effect of powder size and adhesion between HS and the PLA matrix. In addition, they reported interesting results for scaled-up composites.

Pradhan et al. [31] studied the incorporation of walnut shell powder. The presence of these particles modified the density and porosity of the composites, along with their tensile, compressive, flexural, and impact strengths.

In their work, Kufel et al. [32] studied the mechanical and physical properties of hybrid composites. They added 10%, 15%, and 20% by weight basalt fibers (BFs) and ground hazelnut shells (HSs) to the polypropylene matrix.

On the basis of the above considerations, the present work focused on the manufacture and characterization of biodegradable Mater-Bi-based biocomposites filled with HS. The obtained biocomposites were characterized to study the rheology, static and dynamic mechanical properties, and thermal and morphological properties to determine the role of hazelnut shells in the Mater-Bi matrix. In addition, both biocomposites were molded using an industrial injection molding machine to evaluate the scale-up feasibility and mechanical properties of the scaled-up biocomposites.

## 2. Materials and Methods

### 2.1. Materials

The material used in this study was a biodegradable polyester Mater-Bi^®^ El51N0 supplied by Novamont (Novamont, Novara, Italy).

Mater-Bi^®^ has a proprietary composition [33]; however, considering the information available in the literature, it is a class of starch-based materials.

Table 1 shows some relevant properties of the polymer used.

Hazelnut shell (HS) powder was provided by Agrindustria Tecco (Agrindustria Tecco, Cuneo, Italy) [30].

Figure 1 shows that the predominant particle size range was 120–240 μm.

### 2.2. Manufacturing of Biocomposites

An OMC coronating twin-screw extruder (OMC, Saronno, Italy) with a screw diameter of 19 mm and a length-to-diameter ratio of 35 mm was used to prepare the Mater-Bi-based composites. The filler concentrations used were 10% and 30% by weight. The temperature profile used was 150-160-170-170-170-180-180 °C (die), the screw speed was set to 160 rpm, and the gravity feeder was set to 10 rpm. To prevent Mater-Bi from hydrolyzing and breaking the ester bond during processing [34], Mater-Bi and HS were vacuum dried at 60 °C for 4 h and 80 °C overnight, respectively.

#### 2.2.1. Compression Molding (CM)

Compression molding samples were prepared in a Carver (Carver, Wabash, IN, USA) laboratory hydraulic press at a temperature of 180 °C under a mold pressure of 300 psi for about 3 min. Figure 2 illustrates a schematic representation of biocomposite fabrication by compression molding.

#### 2.2.2. Injection Molding (IM)

Biocomposites prepared in the same manner as described above (coronating twin-screw extruder) were injection-molded using a Negri Bossi EOS-65 industrial injection molding machine (Negri Bossi, ColognoMonzese, Italy); see Figure 2. The injection temperature profile was established on the basis of previous work [35]. That is, from die to hopper, the temperatures for the MB matrix were 190, 180, 170, and 170 °C; for the biocomposites of MB, the temperatures were 200, 190, 180, and 180 °C.

Figure 3 shows the samples obtained after injection molding.

### 2.3. Characterization of Biocomposites

The thermal properties of the materials were evaluated by differential scanning calorimetry (DSC). DSC measurements were performed by a DSC-131 Setaram (Setaram, Hillsborough Township, NJ, USA) under a nitrogen gas atmosphere using 10 ± 2 mg of sample and a 10 °C/min heating rate up to 200 °C/min.

The rheological characterization was performed using an ARES G2 (TA Instruments, New Castle, DE, USA) equipped with a parallel-plate geometry (25 mm diameter). The measurements were taken at 180 °C in an angular frequency range of 0.1 to 100 rad/s.

Mater-Bi-based composite samples for rheological tests were prepared via compression molding in a Carver (Carver, Wabash, IN, USA) laboratory hydraulic press. Before testing, all the samples were left to dry under vacuum for 4 h at 60 °C.

Dynamic mechanical thermal analysis was performed using a Metravib DMA 50 (Metravib, Limonest, France). The measurements were carried out at a constant frequency of ω = 1 Hz in a temperature range of 25 to 80 °C with a heating rate of 3 °C/min. Analyses were carried out on samples cut to the size of 10 × 30 × ≈ 0.6 mm before being mounted in the DMTA apparatus. Before testing, all the samples were left to dry under vacuum for 4 h at 60 °C.

The mechanical property tests were performed according to ASTM D638-14 [36]. The tests were performed in the universal testing machine Instron mod. 3365 (Instron, High Wycombe, UK) with a load cell of 5 kN and a crosshead rate of 100 mm/min.

The notched Izod impact resistance was determined as specified by the ASTM D256 [37].

The morphology of HS powder was analyzed by scanning electron microscopy (SEM), performed by means of a Phenom Pro X microscope (Phenom-World, Eindhoven, The Netherlands), whereas SEM images of the biocomposites were obtained with a Quanta 200F scanning electron microscope (FEI Co., Hillsboro, OR, USA). Prior to SEM examination, the sheets were fractured in liquid nitrogen and gilded to make them conductive.

## 3. Results and Discussion

DSC scans of MB and biocomposites were carried out in order to characterize the thermal properties of these developed materials. The DSC curves of the biocomposites with regard to the matrix are shown in Figure 4, and the DSC parameters are summarized in Table 2.

Since MB is the only component, the enthalpies were normalized by dividing the corresponding enthalpy value (peak integral) by the weight fraction of MB in the sample.

From the normalized enthalpy values, the degree of crystallinity was found to increase with increasing HS filler content. This result has been found previously [28,38] and was attributed to a slight nucleating action of the HS particles promoting crystallization.

The presence of lignin in hazelnut shells promotes the formation of crystallites. In fact, the addition of HS to the matrix causes an increase in crystallinity thanks to the action of lignin, which acts as a nucleating agent [28].

In their work, Balart et al. [28] noted the presence of this effect. The presence of lignin, which acts as a nucleating agent, as mentioned above, increases the crystallinity of the biocomposite.

Salazar-Cruz et al. [39] obtained similar results. In fact, by adding filler content, they obtained an increase in the composite’s melting temperature and ΔHm, indicating that a larger number of crystalline zones were present. In addition, in this case, the filler acts as a nucleating agent ordering the polymer chains, which was reflected as an increase in the melting temperature.

In Figure 5, the flow curves of MB and MB/HS biocomposites as a function of angular frequency are reported. The viscosity curves clearly show that HS significantly affected the response of the biocomposites. In particular, the flow curves of the biocomposites do not show any Newtonian plateau at a low frequency, and typical shear-thinning [40] behavior with the viscosity dramatically increasing with decreasing the frequency was observed. Moreover, the flow curves of the biocomposites showed a more pronounced non-Newtonian behavior. Because of their higher viscosity, the non-Newtonian behavior of the biocomposites was more pronounced with an increased content of filler and increased frequency. The flow curves of the two biocomposites are very similar; because of its more pronounced non-Newtonian behavior, the viscosity of the biocomposite with 30% HS became lower than that of the biocomposite with a lower content of HS.

Figure 6 shows the storage modulus as a function of angular frequency for all the investigated samples. It can be seen that the addition of HS increased the storage modulus, and the storage modulus increased with increasing the filler concentration. More specifically, at low frequencies, the storage modulus values of the two biocomposites were almost independent of frequency, thus indicating that the viscoelastic behavior of the melt turns to a solid-like behavior [41,42]. This behavior was more pronounced with an increased HS content. At a higher frequency, the storage moduli of the biocomposites and the pure matrix became very similar both in value and slope. This means that at low frequencies and high relaxation times, the behavior of the biocomposite is determined by the presence of the inert filler, while at high frequencies and low relaxation times, the viscoelastic behavior is governed by the matrix.

Figure 7 shows the variation in E′ with temperature for all the investigated samples.

It was observed that the E′ of the biocomposites increased when increasing the content of HS throughout the investigated temperature range because of the higher stiffness due to the presence of the hard inert filler. In more detail, MB, MB/HS 10%, and MB/HS 30% showed storage modulus values of 1454, 1596, and 1898 MPa, respectively, at room temperature.

At higher temperatures, the difference in the moduli between the matrix and the biocomposites increased and became very remarkable at temperatures higher than the glass transition temperatures. At these temperatures, the modulus of the matrix fell, while the modulus of the two biocomposites remained almost constant. The value of the storage modulus of the pure matrix at 60 °C (538 MPa) was the same as that of the biocomposite at about 80 °C.

The effect of the content of the filler disappeared when increasing the temperature, and at temperatures higher than 50 °C, the E’ curves of the biocomposite samples were almost superimposed.

The corresponding tan δ curves reported in Figure 8 show the typical behavior of the polymeric systems with low values at low temperatures, a sudden increase with increasing the temperature, and a maximum at the glass transition temperature. The Tg of the pure matrix was about 63 °C, and the value of tan δ was almost 1. However, the shape and values of tan δ were strongly modified by the presence of the inert filler. Indeed, when increasing the temperature, the rise in the tan δ curves of the biocomposites was very low, and the maximum value did not increase significantly with respect to the value at room temperature.

Moreover, the shape of the curves was different with respect to that of the matrix because the curve was very broad, and the maximum was not observed in the investigated temperature range. This means that the glass transition temperature shifted toward a higher temperature, as has been observed for other filled systems [43]. The dramatic decrease in the tan δ curves of the biocomposites reflects the solid-like nature of the biocomposite above the glass transition temperature, as evidenced by the G’ curves, Figure 6, and the lower viscous loss during the transition shown by the very small area under the peak of the glass transition.

The effects of HS on tensile strength and elongation at break are shown in Figure 9. The presence of HS led to an increase in the tensile strength of the MB/HS 10% sample by about 10% compared with that of MB. On the contrary, MB/HS 30% showed a decrease in tensile strength of about 48% compared with that of MB.

This decrease in tensile strength of the MB/HS 30% sample may be associated with premature failure of the samples; see Figure 10.

Indeed, the HS provoked a decrease in the elongation at break of the polymer matrix. In more detail, the elongation at break of MB was 2.0%, while MB/HS 10% and 30% showed elongation at break values of about 1.7% and 0.6%, respectively.

Figure 11 shows the SEM micrographs of the fracture surfaces of the two biocomposite materials. The first feature observed was that many particles were pulled out of the matrix, leaving small holes. In addition, the lack of a strong interaction between the HS and the MB matrix was clearly observable, suggesting poor adhesion at the particle–polymer interface.

Of course, this lack of particle–polymer continuity, in particular for the sample with a higher content of HS, is responsible for the stress concentrations and poor transfer of particle loads to the matrix. This explains why the tensile strength of the MB/HS composite at 30% was about 50% lower. High magnification micrographs provide further direct evidence of irregular shapes, which can lead to poor biocomposite strength.

Figure 12 shows the stress–strain curves of the injection-molded specimens compared with those of the compression-molded specimens.

As shown in Figure 12, the elongation at break of the injection-molded specimens was slightly lower than that of the compression-molded specimens for both samples. In addition, both injection-molded specimens showed a significant increase in the elastic modulus but a slight decrease in the tensile strength due to the lower value of the elongation at break; see Table 3.

It should be noted that the comparison with 30% was not reported because the injection molding of the sample with 30% filler was very difficult because of both the high viscosity and the size of some agglomeration of the inert particles.

Table 4 shows the results of the impact tests performed on the injection-molded specimens.

The impact resistance increased with the addition of filler.

## 4. Conclusions

HS can be considered a good candidate as a filler of biodegradable polymers. Indeed, at a moderate concentration, HS improves the rigidity, the impact strength, and, in particular, the thermomechanical resistance of the polymer matrix. In particular, the elastic modulus of the biocomposite increased by more than 25% for the compression-molded samples and more than 40% for the injection-molded samples. The tensile strength was slightly increased without any significant loss in the elongation at break. A remarkable improvement in the impact strength of the injection-molded samples, more than 30%, was also observed.

As expected, because of the increase in viscosity and some agglomeration of the particles of the inert fillers, the processability in injection molding was certainly worsened, but at a concentration of 10% wt/wt, the biocomposite was easily processable.

In summary, the results of this study demonstrate that the fabrication of Mater-Bi-based composites using HS represents huge potential for its practical use in several industrial applications.

## Figures and Tables

**Figure 1 polymers-14-02151-f001:**
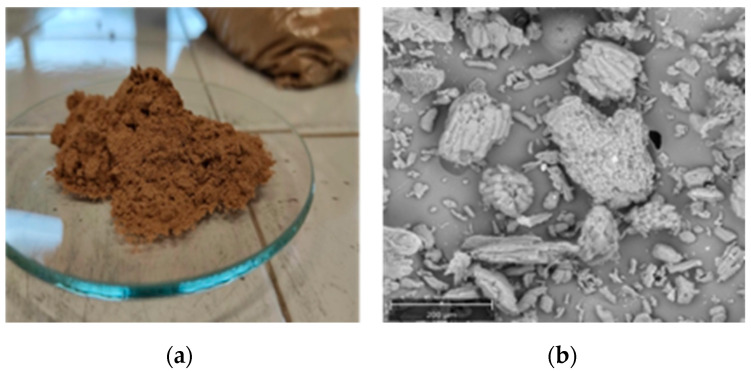
(**a**) HS powder; (**b**) scanning electronic microscope (SEM) image of HS (scale marked as 200 μm).

**Figure 2 polymers-14-02151-f002:**
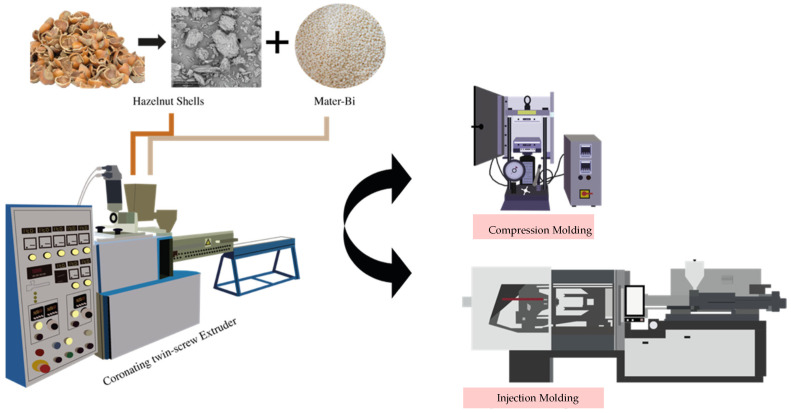
Schematic for biocomposite fabrication by compression molding and injection molding.

**Figure 3 polymers-14-02151-f003:**
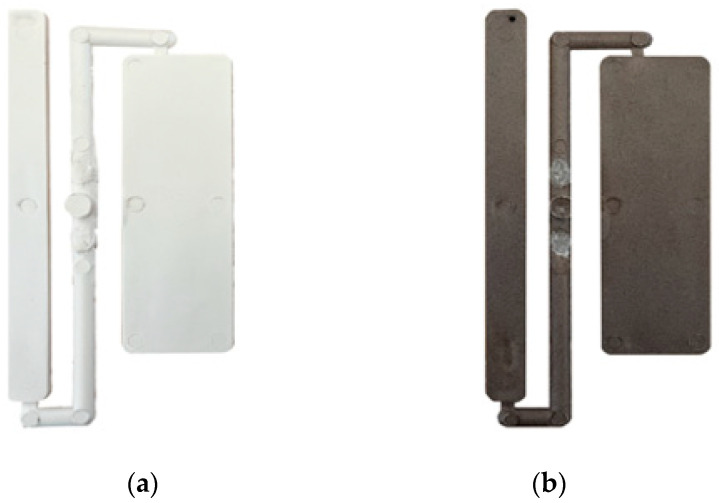
Injection molding samples: (**a**) MB; (**b**) MB/HS 10%.

**Figure 4 polymers-14-02151-f004:**
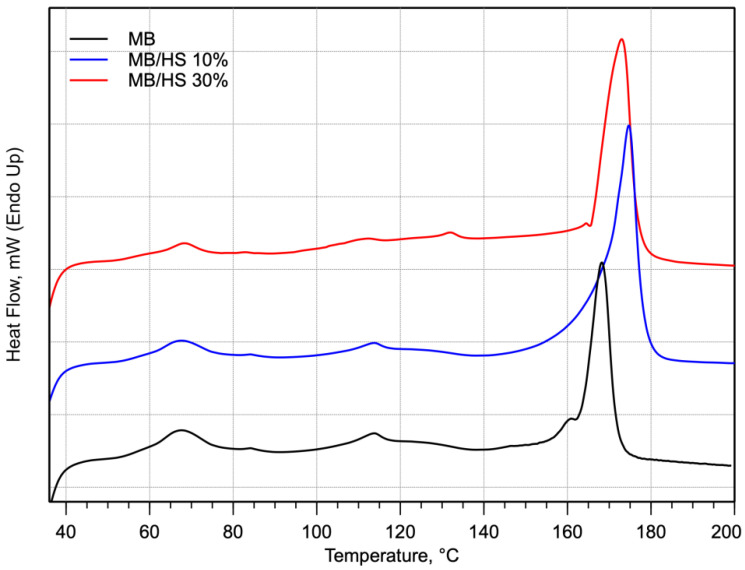
DSC thermograms recorded during the first heating for MB and biocomposites.

**Figure 5 polymers-14-02151-f005:**
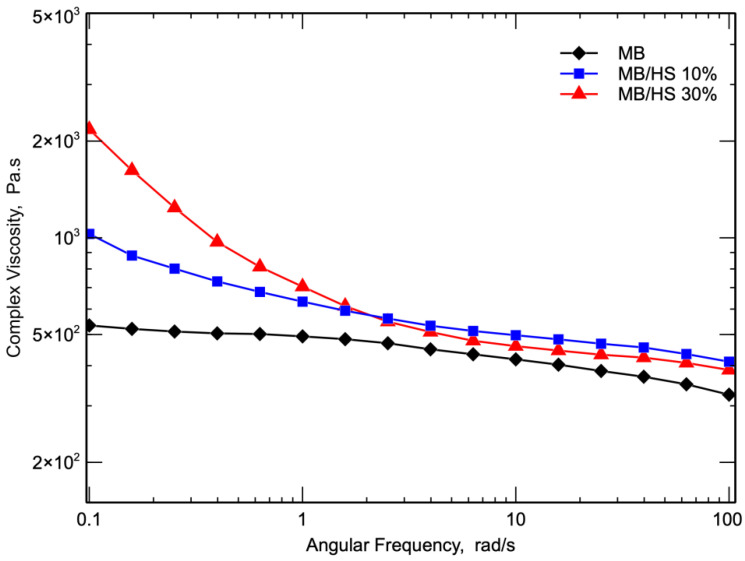
Complex viscosity as a function of frequency of MB and biocomposites.

**Figure 6 polymers-14-02151-f006:**
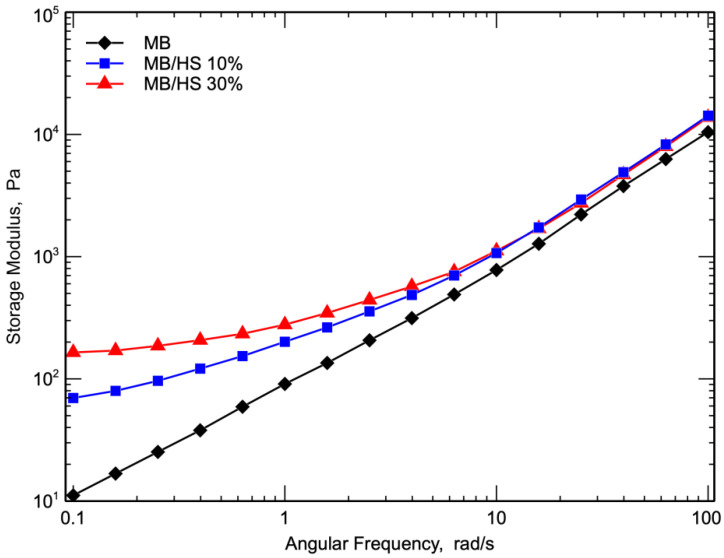
Storage modulus as a function of frequency of MB and biocomposites.

**Figure 7 polymers-14-02151-f007:**
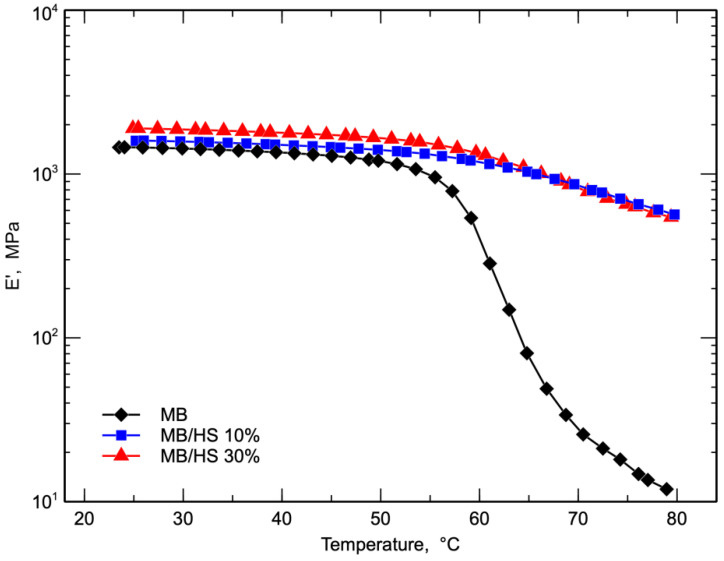
DMA storage modulus (E′) curves of MB and biocomposites.

**Figure 8 polymers-14-02151-f008:**
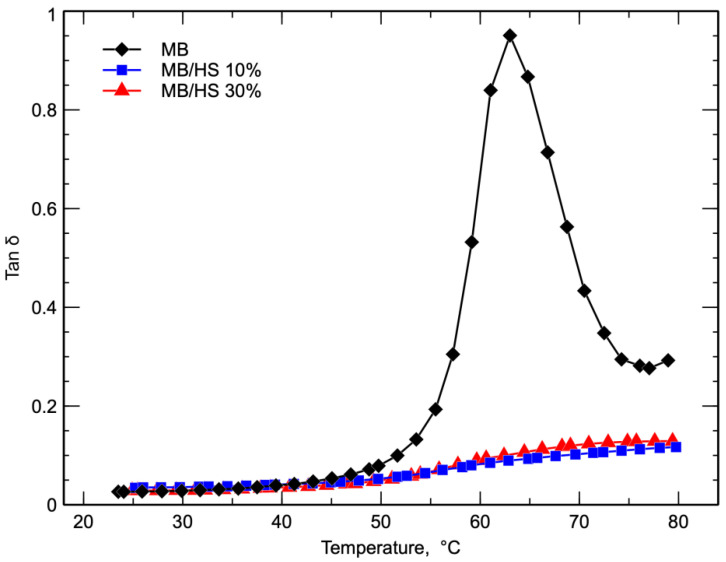
Tan δ curves of MB and biocomposites.

**Figure 9 polymers-14-02151-f009:**
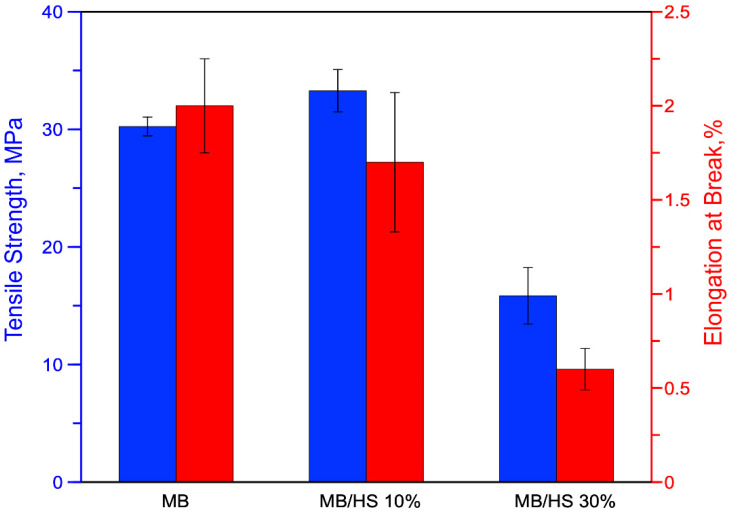
Histogram of tensile strength and elongation at break of MB and biocomposite.

**Figure 10 polymers-14-02151-f010:**
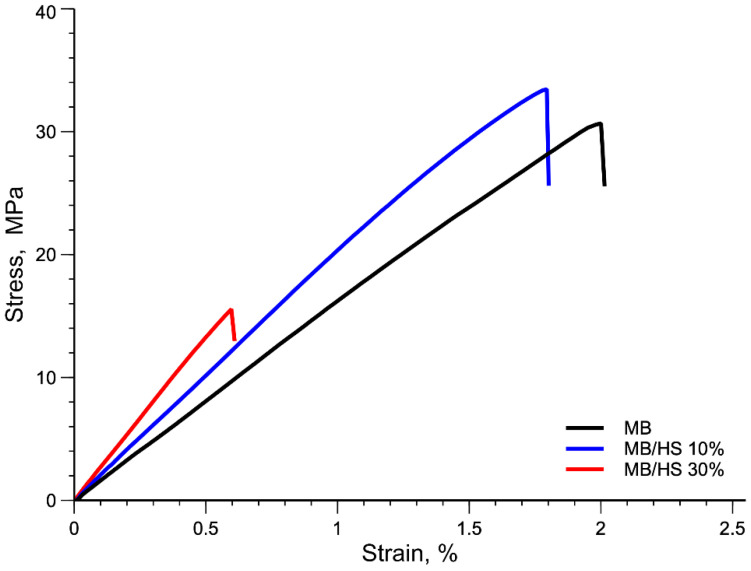
Stress–strain curve of MB and biocomposites.

**Figure 11 polymers-14-02151-f011:**
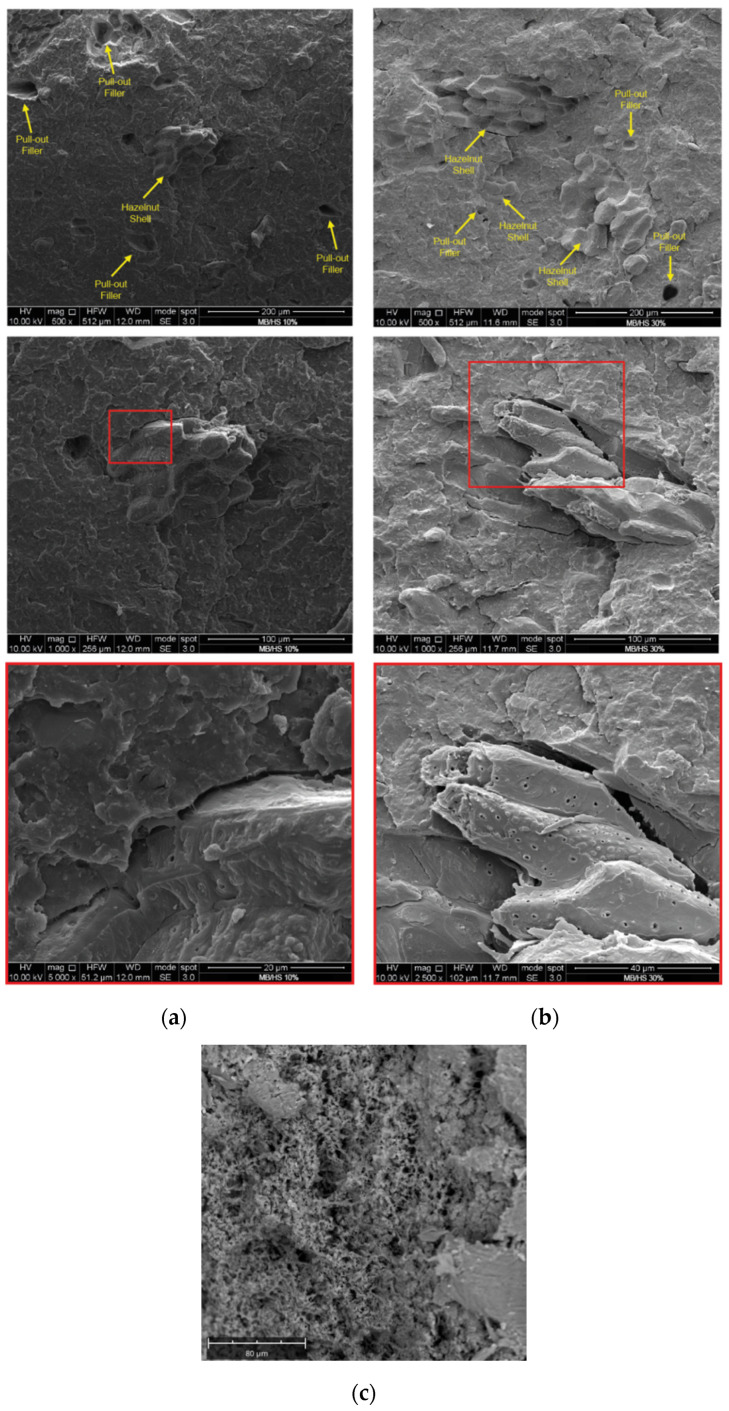
SEM micrographs of the two biocomposites, (**a**) MB/HS 10% and (**b**) MB/HS 30%; (**c**) MB.

**Figure 12 polymers-14-02151-f012:**
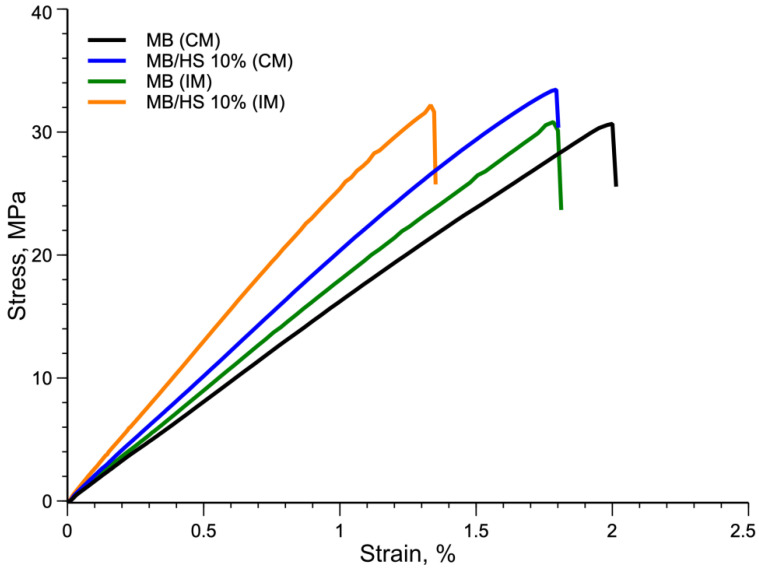
Compression- and injection-molded samples’ stress–strain curves.

**Table 1 polymers-14-02151-t001:** Relevant properties of the polymer investigated.

Polymer	Density, g/cm^3^	MFR, g/10 min (2.16 kg at 230 °C)	Melting Point, °C
Mater-Bi^®^ El51N0	1.23	19	167

**Table 2 polymers-14-02151-t002:** DSC first heating results for MB and biocomposites.

Sample Name	Tm, °C	ΔHm(J/g)
MB	168.5	27.1
MB/HS 10%	175.2	31.5
MB/HS 30%	173.1	39.8

Tm, melting temperature; ΔHm, melting enthalpy.

**Table 3 polymers-14-02151-t003:** Elastic modulus, tensile strength, and elongation at break of compression-molded and injection-molded MB and MB/HZ 10% samples.

Sample Name	Elastic Modulus, MPa	Tensile Strength, MPa	Elongation at Break, %
MB (CM)	1618 ± 22	30.6 ± 1.1	2.0 ± 0.25
MB/HS 10% (CM)	2039 ± 27	33.5 ± 1.6	1.7 ± 0.35
MB (IM)	1798 ± 19	29.7 ± 1.4	1.8 ± 0.15
MB/HS 10% (IM)	2595 ± 37	30.7 ± 1.2	1.3 ± 0.05

**Table 4 polymers-14-02151-t004:** Impact strength.

Property	MB	MB/HS 10%
Impact strength, KJ/m^2^	21.7 ± 1.3	28.8 ± 2.7

## Data Availability

The data presented in this study are available on request from the corresponding author.

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
