# Peer review of "The Use of Waste Hazelnut Shells as a Reinforcement in the Development of Green Biocomposites"

_polymers, 2022, doi:10.3390/polym14112151_

Round 1

Reviewer 1 Report

After review the manuscript, this is an interesting work, but there are several recommendations/observations that need to be corrected, which are listed following:

-Please homogenize if use the abbreviation HS or Hazelnut shells in whole the manuscript. Also, first time an abbreviation is written must be defined.

.It would be interesting to know the hazelnut shells compositions, is it a lingnocellulosic material? which is the lignin, cellulose, hemicellulose content? there are several works who report this content and relate the content of them I behavior of thermal, and mechanical properties.

.In line 75 indicate that composition of Mater-bi was obtained from literature, but if material is commercial the thecnical sheet must be provided for manufacturer.

-It is confusing that in line 79 indicate that particle size is 90-150 micrometers, and in figure 1 report a particle size 120-240 micrometers, and please be more specific shot how the particle size was determinate by means SEM.

-based on what decide to use 10 and 30% HS in bio composites?

-In figure 3 there are two samples obtained from injection that can be use for analysis, please indicate which of them was used.

-For DSC analysis please indicate which was the initial temperature. Also, it is common that when this kind of analysis is carried out, if polymer is processed (extrusion, injection, mixing, etc) a double scan is carried out in the aim to delete the thermal history of material, specially if a combination of these process were used for polymer, so I recommend to carry out a double scan in DSC to delete the thermal history.

-For mechanical tests, please indicate how many replies were carrie out for each biocomposite.

-IN table 2 report a Tg value but in DSC thermogram, at this temperature it seem to be a peak not a step, as usually the Tg is identified, can explain this? What about other events that are visible in thermogram, at 115ºC and for bio composites around 135ºC? Please explain them.

-A more deep explanation about the nucleating effect of HS in bio composites must be done, for these the authors can check next references: 

https://doi.org/10.3390/molecules26195927

https://doi.org/10.3390/molecules27020426

https://doi.org/10.1002/app.1993.070500709

-What is it mean that complex viscosity curve cross at angular frequency of 2 rads/s and bio composite HS 10% shows a higher value?

-I recommend to make a deep discussion about the rheological (fig 5-8) behavior of G`and E`, I mean only a description of behavior in figures is done but not justify the results, what kind of interaction between polymer matrix and HS happened to increase the storage modulus or complex viscosity for instance? 

-Why the peak in Tan delta curve is higher in MB than bio composites? I mean, the presence of HS is responsable for this? and explain why.

-When discuss SEM images indicate that there is a poor adhesion between HS particles and polymer matrix, so how can explain then that HS particles act as nucleating agent without  a good adhesion? Also I recommend to include the MB pure SEM image for a better comparison between pure polymer and bio composite, furthermore to use the same magnification in images for a good comparison.

In general is an interesting work, but it is necessary to make a deep discussion of results.

Reviewer 2 Report

Review report

The authors reported work on reinforcement of green Biocomposites using hazelnut powders as a filler. The effects of reinforcement on the morphology, static and dynamic mechanical properties, and thermal and rheological properties of MB/HS Biocomposites were presented. The paper is well presented and should be accepted after a few revisions.

Following are my comments.

  1. Pg 1 Line 36: change the word “poorer” with “inferior”
  2. Pg 2 Line 52 to 63: improve the quality of literature review with other relevant studies of hazelnut shell powder irrespective of the usage in Biocomposites. It will hence the information for the readers and will provide them with a better understanding of the research gap addressed in this study. A few examples of the related studies are mentioned below:
    1. https://doi.org/10.3390/ma14061368
    2. doi: 3390/polym12010018
    3. https://doi.org/10.1177/09673911211020717
  3. Pg 3 Figure 2. Labels are awfully hard to read, please provide a high-resolution image.
  4. Pg 5 In DSC characterization percentage crystallinity change should be measured from the enthalpy values.
  5. The result and discussion section lack any information on the physicochemical characterization of the prepared composites. Data obtained from XRD, FTIR, and XPS characterizations of the prepared composites will further enhance the quality of the presented work. Furthermore, I do understand the current inconvenient situation due to the COVID resurgence. So, from the above-mentioned characterization, any one or two will be enough if not all possible.
  6. The conclusion is too generic. Please provide the information obtained from the results in numerical form and the same goes for the abstract.

Round 2

Reviewer 1 Report

After review the corrected version, most of observations/recommendations were taken in account and the manuscript shows a significant improve, the only one detail that need to be corrected is the references format, due they are not according with Authors instructions.

Author Response

We would thank the reviewer for her/his valuables comments.

The required corrections have been made

Reviewer 2 Report

The authors have sufficiently improved the quality of the manuscript and it can be published in its current form.

Author Response

thanks for your valuable suggestions